# Risk and outcomes of healthcare-associated infections in three hospitals in Bobo Dioulasso, Burkina Faso, 2022: A longitudinal study

**Arsène Hema** [1]*, **Satouro Arsène Somé**[2], **Odilon Kaboré**[3], **Soufiane Sanou**[2], **Armel Poda**[4,5], **Ziemlé Clément Meda**[5,6], **Abdoul Salam Ouedraogo**[3,5], **Léon Savadogo**[5]

**1** Quality Department of Sourô Sanou Teaching Hospital, Bobo-Dioulasso, Burkina Faso, **2** Centre Muraz, National Public Health Institute, Bobo-Dioulasso, Burkina Faso, **3** Bacteriology and virology laboratory, Bobo-Dioulasso, Burkina Faso, **4** Infectious Diseases Department of Sourô Sanou Teaching Hospital, Bobo-Dioulasso, Burkina Faso, **5** National Institute of Health Sciences, Nazi Boni University, Bobo-Dioulasso, Burkina Faso, **6** Research and Epidemiology Department of Sourô Sanou Teaching Hospital, Bobo-Dioulasso, Burkina

* hemaarsene@yahoo.fr

## Abstract

### Background

Healthcare-associated infections (HAIs) are one of the most common adverse events in healthcare and represent a major public health problem. The present study was conducted to analyze the incidence, risk factors, and outcomes of HAIs through active surveillance in three hospitals in the city of Bobo Dioulasso, Burkina Faso.

### Methods

A prospective, longitudinal, multicenter study was conducted from May 1 to November 30, 2022, in two district hospitals (Do and Dafra) and the Sourô Sanou Teaching Hospital (CHUSS), Burkina Faso. Consenting patients hospitalized for reasons other than infection, cancer, immunosuppression in the postoperative care ward of Do or of Dafra district hospitals, intensive care unit (ICU)/CHUSS, neonatal ward/CHUSS, and gynecology and obstetrics postoperative care ward/CHUSS during a 2-month inclusion period in district hospitals and 4 months for CHUSS wards. For this study, we used the operational definitions of the French Technical Committee for Nosocomial Infections and Healthcare-associated Infections, with slight modifications. Logistic regression was used to analyze risk factors of HAIs.

### Results

Of the 664 patients enrolled, 166 experienced an HAI, with a cumulative incidence rate of 25% (95%CI: 21.7%–28.3%) or an incidence density rate of 36.7 per 1000 patient-days (95% CI: 31.7–42.9). Surgical site infections (SSI) (44%), followed by neonatal infections (42%) were the most common HAIs. Enterobacterales represented 60% of the bacteria identified in

**Data availability statement:** All relevant data are within the manuscript and its Supporting Information files.

**Funding:** The author(s) received no specific funding for this work.;

**Competing interests:** The authors have declared that no competing interests exist.

HAIs, and 38.9% of them were extended spectrum β-lactamase (ESBL) producers. Factors associated with HAIs were admission in the neonatal ward (aOR = 6.6; 95%CI:1.1–40.2), ICU (aOR = 3.3; 95%CI:1.3–8.5), previous hospital stay longer than two days (aOR = 2.0; 95%CI:1.2–3.3), or male sex (aOR = 1.8; 95%CI:1.1–3.0). In addition, HAIs were associated with longer follow-up, hospitalization, and mortality (18.1%; 95% 95%CI:12.1–24.4). Deaths were only recorded in the ICU and neonatal ward, with case fatality rates of 45.4% (95% 95%CI: 27.5–63.4) and 21.4% (95% 95%CI: 11.6–31.3), respectively, p = 0.019.

## Conclusions

The incidence of HAIs was relatively high in the three hospitals in Bobo Dioulasso, Burkina Faso. It is essential to implement rigorous protocols for patient management, to reduce the incidence of HAIs and the spread of resistant pathogens in general and Enterobacterales in particular.

## Introduction

Healthcare-associated infections (HAIs) are one of the most common adverse events in healthcare and a major public health problem in both developed and resource-limited countries [1]. HAIs are associated with increased healthcare costs, prolonged hospital stays, functional disability, and patient death [1,2]. Most HAIs are caused by bacteria, viruses or microscopic fungi that become multi-resistant due to inappropriate use of antimicrobials [3]. Thus, multi-resistant microorganisms cause infections with limited therapeutic options, leading to longer hospital stays and higher mortality rates compared to HAIs caused by susceptible bacteria of the same species [2,4,5]. Therefore, the fight against HAIs is inextricably linked to the fight against antimicrobial resistance [1,6].Many factors promote HAIs, including healthcare systems and procedures, human behavior influenced by education, socioeconomic constraints, and often societal norms and beliefs. However, 30% of HAIs are considered preventable through adherence to hospital hygiene measures [7].

According to the World Health Organization (WHO), HAIs affect 1.4 million people worldwide at any given time, with 3.5% to 12% in developed countries (9% in the United Kingdom, 4.4% in France, and 6.9% in Belgium) compared to 25% in low- and middle-income countries [1,2]. In Africa, hospital-wide prevalence of HAIs varied from 6.7% to 18.7%, with a high prevalence that can exceed 50% in ICUs [8–12]. Most of these studies are prevalence studies, which have the advantage of being easy and quick to conduct, but the disadvantage of low precision and lack of information on the outcomes of HAIs [13].

In Burkina Faso, data on the incidence of HAIs are scarce due to the lack of an HAI surveillance system [14,15]. In addition, hospital overcrowding, limited resources, structural and logistical challenges, combined with inadequate training of professionals in hospital hygiene, antimicrobial resistance, and their inadequate monitoring, suggest a high risk of HAIs. The present study aims to analyze the extent, riskfactors and outcomes of associated infections through active surveillance in three hospitals inthe city of Bobo Dioulasso, Burkina Faso.

## Methods

### Study design and setting

This prospective, longitudinal, multicenter study was conducted from May 1 to November 30, 2022, in 3 health care facilities in Bobo-Dioulasso, the second largest city in Burkina Faso. The facilities included two district hospitals (Do and Dafra) and one teaching hospital (Sourô

Sanou Teaching Hospital (CHUSS) in Bobo-Dioulasso. In the district hospitals, the study was conducted in the postoperative care wards. In the Sourô Sanou teaching hospital, the study was conducted in the intensive care unit, the gynecology and obstetrics postoperative care ward and the neonatal ward.

## Postoperative care ward of Dafra district hospital

The post-operative ward of Dafra District Hospital received 888 post-operative patients in 2022. This ward consists of two inpatient rooms with a total capacity of 16 beds. During the survey, the service was staffed by 52 nurses, two hospital hygiene technicians, and two permanent physicians.

## Postoperative care ward of Do district hospital

The postoperative ward of Do District Hospital received 1706 postoperative patients in 2022. This ward consists of six inpatient rooms, with a total capacity of 24 beds. At the time of the survey, the service was staffed by 59 nurses, 1 hospital hygiene technician, five permanent physicians, and approximately five medical trainees.

## Intensive Care Units(ICU) of Sourô Sanou teaching hospital

It is the reference center for intensive care of four health regions out of 13 of Burkina Faso. In 2022, 525 patients were admitted to the intensive care unit of Sourô Sanou Teaching Hospital. This unit has 14 inpatient beds. The service is staffed by five Physicians Anaesthetist resuscitators, approximately 10 physicians' residents, 23 nurses and three (03) hospital hygiene technicians

## Neonatalward of Sourô Sanou teaching hospital

It was the only neonatal ward in western Burkina Faso and a reference center for neonatal care of four health regions.In 2022, the neonatalward of Sourô Sanou Teaching Hospital received 2,765 new-borns. It has a hospital room that includes 6 incubators and 34 cradles. The staff consists of two doctors specializing in pediatrics, two general practitioners, approximately 10 resident physicians, 28 nurses and three hospital hygiene technicians.

## Gynecology and obstetrics postoperative care ward of Sourô Sanou teaching hospital

In 2022, the gynecology and obstetrics postoperative care ward of Sourô Sanou Teaching Hospital treated 1,946 women, 1,433 of whom had cesarean sections. The ward has 30 inpatient beds and a staff of 16 physicians, four nurses, and 21 midwives.

## Study population

The population of this study consisted of the cohort of all patients admitted to the neonatal ward, ICU or the gynecology and obstetrics postoperative care ward of Sourô Sanou Teaching Hospital, as well as the postoperative care ward of Do District Hospital and of Dafra District Hospital during the period of active HAI surveillance.

## Inclusion criteria

Were included in this study, patients meeting the following conditions:

- Admitted to the postoperative care ward of Do or Dafra District Hospital, from May 1 to June 30, 2022, or

- Admitted to ICU or neonatal ward of Sourô Sanou Teaching Hospital, from May 14 to August 14, 2022, or

- Admitted to the gynecology and obstetrics postoperative care ward of Sourô Sanou Teaching Hospital, from August 1 to November 30, 2022 and have given their consent

### Exclusion criteria

Patients were not included in this study if they met any of the following criteria:

- Patients who have been hospitalized for less than 48 hours in the study ward,

- Had a primary diagnosis of sepsis, cancer, infection, or immunocompromised state.

### Surveillance and data collection

Based on data from a previous prevalence study [15], we targeted 4 infections to monitor for this study: urinary infections, systemic infections, surgical site infections (SSI) and neonatal infections. A team composed of an epidemiologist, two clinical doctors and three nursing managers was responsible for active data collection in collaboration with the care unit supervisor. Data were collected from patient records, bandage records, hospitalization records and patient interviews. Case identification was active. For this purpose, the data collection team examined hospitalization registers three times a week to identify eligible patients. Information collected from these sources was supplemented by information from patients and nursing staff. Data was then recorded on individual data collection sheets which included: sociodemographic data (age, sex, origin, profession), risk factors related to care, initial or curative antibiotic therapy, HAI type, pathogen in question, patient outcomes. Surgical patients included were followed until the surgical wound healed, even after discharge from the hospital. The condition of the surgical wounds was monitored during outpatient bandage visits. Non-operated patientswere followed until they left the hospital. In all cases, patient follow-up did not exceed 30 days. In the event of HAI, biological samples when possible were taken when possible, depending on the type of suspected infection (blood, pus, urine) and a bacteriological analysis were performed in the CHUSS laboratory (S1 Appendice).

### Definition of outcome variables

The primary outcome variable was occurrence of infection after 48 h of hospitalization in a patient with nosymptomatic or incubating infection at the time hospital admission. For this study, we used the operational definitions of the French Technical Committee for Nosocomial Infections and Healthcare-Related Infections (CTINILS) [16], with slight modification (S2 Table). This committee largely adopts the definitions of the Center of Disease Control and prevention (CDC).

### Variables

**Outcome variables.** Outcome variables were HAI and its subsequent outcome. HAI was defined by the presence of at least one of the infections targeted by surveillance during this study (SSI, urinary tract infection, systemic infection, neonatal infection). For the analysis, we coded: 0 for no HAI and 1 for presence of HAI. HAIs outcomes were categorized by recoveries, deaths, discharges against medical advice and lost to follow-up.

**Predictor variables.**

- Age: categorized into four groups: 0–14 years, 15–35 years, 36–50 years, and >50 years.

- Gender: female and male

- Residence: the patient's place of residence, categorized as urban or rural

- Exposure: Surgery, urinary catheter, peripheral venous catheter, length of stay in hospital before inclusion in the study (in days)

- Ward/Unit: these are the wards/units from which patients were recruited. There were three Wards/Units of Sourô Sanou teaching Hospital (ICU/CHUSS, Neonatology/CHUSS Gyn & Obs/CHUSS), one ward of the district hospital of Dafra (Surgical/Dafra) and one of the district hospital of Do (Surgical/Do).

- Follow-up duration: period between inclusion in the study and HAIs outcome occurrence. The duration of follow-up does not exceed 30 days.

- Previous hospital stay: the time spent in another ward in the same hospital or in another hospital before admission to the study wards.

## Statistical analysis

Data collected on survey forms were entered into Epidata® software version 3.1 and analyzed using Stata® software version 13(S3 file). Patient characteristics, HAI exposure, and ward distribution were described for patients with and without HAIs. Cumulative incidence rates and incidence densities of HAIs and their 95% confidence intervals (CI) were estimated. Bacteriological profile of HAIs was determined and HAI outcomes were described by HAI exposure and unit. Qualitative variables were described by their percentages and quantitative variables by their mean (me) and standard deviation (SD). Proportions were compared using Pearson's chi2 test or Fisher's exact test when appropriate. Means were compared using Student's t-test. Logistic regression (univariate and multivariate) was used to identify the risk factor for HAI. In the univariable logistic regression analysis, patient characteristics (age, sex, location HAI exposures and ward/unit) associated with HCAI at the 20% threshold were selected for multivariable logistic regression. The p-value significance level was 5% for all statistical analyses

## Ethical aspects

This study was conducted as part of the "Surveillance and prevention strategies for healthcare associated infections in Bobo-Dioulasso, Burkina Faso" study approved by the National Ethics Committee of Burkina Faso; approval reference letter number 2022-02-020. Signed informed consent was obtained from the participants and an information sheet was given to each participant. We obtained informed consent, written and signed from parents or guardians for minors. The data collected were anonymized to ensure confidentiality.

## Results

### Description of the study population

A total of 664 patients were included with 471 (70.9%) at the Sourô Sanou Teaching Hospital, 85 (12.8%) at the CMA of Dafra and 108 (16.3%) at the CMA of Do. The study population was mainly female(80.1%) with a mean age of the study population of 21.6 years (sd: 14.3).Neonates (less than 1 month) represented 25% of our study population. Our patients mainly lived in urban areas (88.7%). Regarding exposure, 69.9% of them had undergone surgery,47.1% had a urinary catheter and 50.0% had a venous catheter (Table 1), and the mean length of previous hospital stay was 2.2 days (sd: 0.4). The majority (34.9%) of the included patients were admitted to the gynecology and obstetrics postoperative ward of the CHUSS.

**Table 1. Characteristics of patients, multicenter longitudinal survey, Bobo-Dioulasso 2022.**

| | All patients n (%) | Patients with HAI n (%) | Patients without HAI n (%) | p-value |
|---|---|---|---|---|
| **All patients** | 664 (100) | 166 (100) | 498 (100) | |
| **Patient characteristics** | | | | |
| **Ages group (years)** | | | | 0.000 |
| • 0–14 | 179 (27.0) | 73 (44.0) | 106 (21.3) | |
| • 15–35 | 404 (60.8) | 72 (43.4) | 332 (66.7) | |
| • 36–50 | 65 (9.8) | 14 (8.4) | 51 (10.2) | |
| • >50 | 16 (2.4) | 7 (4.2) | 9 (1.8) | |
| **Sex** | | | | 0.000 |
| • Male | 132 (19.9) | 64 (38.6) | 68 (13.7) | |
| • Female | 532 (80.1) | 102 (61.4) | 430 (86.3) | |
| **Residence** | | | | 0.395 |
| • Urban | 589 (88.7) | 144 (86.8) | 445 (89.4) | |
| • Rural | 75 (11.3) | 22 (13.2) | 53 (10.6) | |
| **Exposure** | | | | |
| • Surgery | 464 (69.9) | 84 (50.6) | 380 (76.3) | 0.000 |
| • Urinary catheter | 313 (47.1) | 69 (41.6) | 244 (49.0) | 0.106 |
| • Peripheral venous catheter | 332 (50.0) | 73 (44.0) | 259 (52.0) | 0.088 |
| • Previous hospital stay >2 days | 140 (21.1) | 43 (25.9) | 97 (19.5) | 0.099 |
| **Ward/Unit** | | | | 0.000 |
| • Postoperative care ward/Dafra | 85 (12.8) | 15 (9.0) | 70 (14.1) | |
| • postoperative care ward/Do | 108 (16.3) | 13 (7.8) | 95 (19.1) | |
| • Gyn & obstpostoperative care ward/CHUSS | 232 (34.9) | 35 (21.1) | 197 (39.5) | |
| • Neonatal ward/CHUSS | 166 (25.0) | 70 (42.2) | 96 (19.3) | |
| • ICU/CHUSS | 73 (11.0) | 33 (19.9) | 40 (8.0) | |

Gyn & obst = Gynecology & obstetric; ICU= Intensive care unit; CHUSS=Sourô Sanou Teaching Hospital

## Healthcare-associated incidence

Of the 664 patients followed, 166 had at least one HAI, resulting in a cumulative incidence rate of 25% (CI: 21.7%–28.3%) or an incidence density rate of 36.7 per 1000 patient-days (CI: 31.7–4.3). The mean time of the occurrence of these infections was 5 days (SD: 3.5). The highest incidences of HAI were observed in ICU (45.2%; CI: 33.5–56.9%) and theneonatalward (42.2%; CI: 34.6–49.7%) (Table 2).

## Typology of HAIs

Of the 166 HAIs, surgical site infections (SSIs) were the most common, accounting for 44.6% (74/166) of all HAIs and occurring in 16.0% of patients who underwent surgery (74/464). After SSIs, the next most common infections were neonatal infections, which accounted for 42.2% (70/166) of HAIs (Fig 1).

## Bacteriological profile of healthcare-associated infections

A total of 45 microorganisms were identified in 79 samples taken from 166 HAIs. The most common microorganisms found among the 45 were *Escherichia coli* (31.9%), *Klebsiella pneumoniae* (17%), *Acinetobacter baumannii* (12.8%), *Pseudomonas aeruginosa* (8.5%),

**Table 2. Incidence of healthcare-associated infection by ward specialty, multicenter longitudinal survey, Bobo-Dioulasso, 2022.**

| Ward/Unit | Cumulative incidence rate %(95%CI) | Incidence density rate ‰ Patient-days (95%CI) |
|---|---|---|
| All ward/Unit | 25 (21.7%–28.3) | 36.7 (31.7–42.9) |
| Postoperative care ward/Dafra | 17.6 (9.4–25.9) | 13.0 (7.9–21.6) |
| Postoperative care ward/Do | 12.0 (5.8–18.3) | 9.8 (5.7–16.8) |
| Gyn & obst postoperative care ward/CHUSS | 15.1 (10.4–19.7) | 35.2 (25.2–49.0) |
| Neonatal ward/CHUSS | 42.2 (34.6–49.7) | 105.6 (83.5–133.4) |
| ICU/CHUSS | 45.2 (33.5–56.9) | 84.6 (60.2–119.1) |
| P-value | 0.000 | 0.000 |

Gyn & obst = Gynecology & obstetric; ICU= Intensive care unit; CHUSS=Sourô Sanou Teaching Hospital

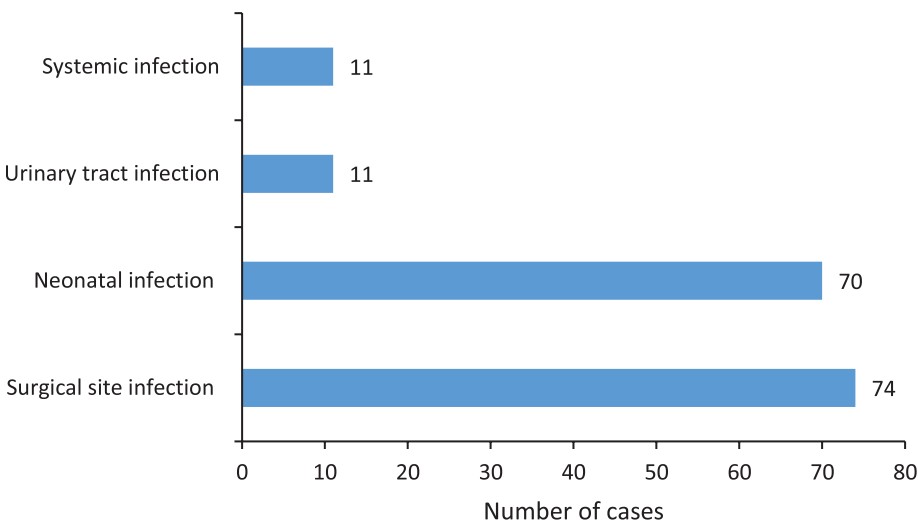

**Fig1. Distribution of healthcare-associated infection(n = 166), multicenter longitudinal survey, Bobo-Dioulasso 2022.**

*Staphylococcus epidermidis* (8.5%), *Staphylococcus aureus* (6.4%), *Enterococcus spp* (4.3%), *Klebsiella oxytoca* (4.3%), *Pantoea spp* (2.1%). Fig 2 summarizes the distribution of different groups of microorganisms. Resistances to 3rd generation cephalosporins, amoxicillin/clavulanate and amoxicillin were 51.8% (14/27), 69.2% (18/26) and 100% (13/13), respectively. The frequencies of resistance to cephalosporins and amoxicillin/clavulanate were highest in the neonatal ward (100% & 100%) and the ICU (66.7% & 100%). The proportion of extended-spectrum beta-lactamase producing Enterobacterales(ESBLE) was 38.9% (7/28). They were isolated in the neonatal ward (n = 2), in the ICU (n = 3) and in the postoperative-ward of Dafra (n = 2). Of three (03) *Staphylococcus aureus* identified, one (01) was methicillin resistant (ICU) 33.3%.

## Factors associated with HAI risk

After multivariate analysis, only previous hospital stay of more than 2 days(aOR = 2.0; CI: 1.2–3.3), male sex (aOR = 1.8; 95%CI: 1.1–3.0), admission in neonatal ward (aOR = 6.6; 95%CI: 1.1–40.2) or ICU (aOR = 3.3; 95%CI: 1.3–8.5) were factors associated with a higher risk of HAI. There was no association between age group and HAIs (Table 3).

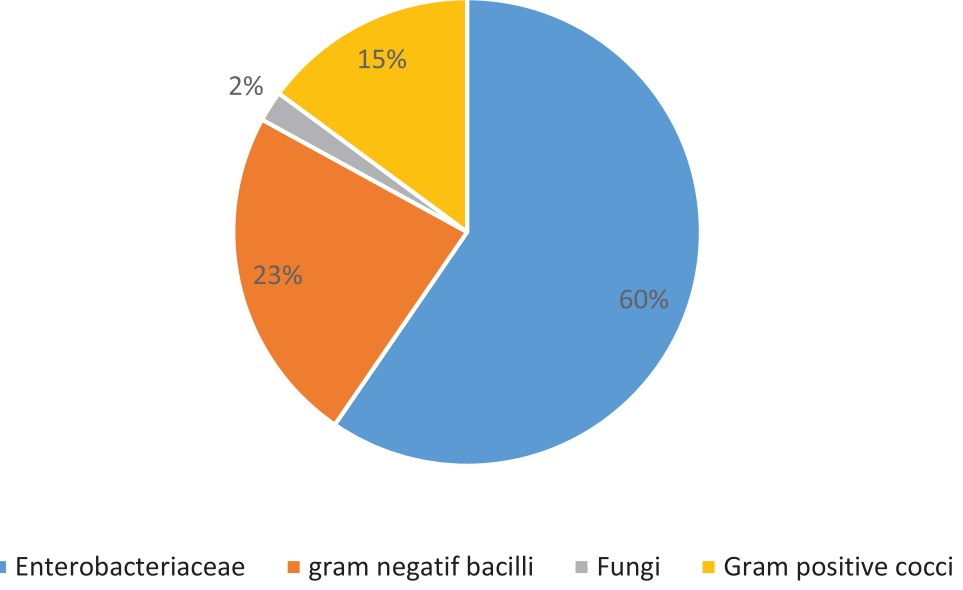

**Fig 2. Groups of microorganisms (n = 45), multicenter longitudinal survey, Bobo-Dioulasso 2022.**

**Table 3. Healthcare-associated infection risk factor, multicenter longitudinal survey, Bobo-Dioulasso, 2022.**

| Variables | cOR [95%CI]. | p-value* | aOR [95%CI]. | p-value* |
|---|---|---|---|---|
| **Ages group (years)** | | 0.000 | | 0.690 |
| • 0–14 | 1 | | 1 | |
| • 15–35 | 0.3 [0.2–0.5] | | 1.9 [0.6–6.5] | |
| • 36–50 | 0.4 [0.2–0.8] | | 1.7 [0.5–6.3] | |
| • >50 | 1.1 [0.4–3.2] | | 1.3 [0.3–6.0] | |
| **Gender** | | 0.000 | | |
| • Female | 1 | | 1 | **0.036** |
| • Male | 4.0 [2.6–5.9] | | 1.8 [1.1–3.0] | |
| **Residence** | | 0.358 | | |
| • Urban | 1 | | | |
| • Rural | 1.3 [0.8–2.2] | | | |
| **Exposure** | | | | |
| • Surgery | 0.3 [0.2–0.5] | 0.000 | 1.4 [0.5–3.6] | 0.473 |
| • Urinary catheter | 0.7 [0.5–1.1] | 0.097 | 1.2 [0.5–2.6] | 0.706 |
| • Peripheral venous catheter | 0.7 [0.5–1.0] | 0.074 | 0.9 [0.3–2.1] | 0.763 |
| • Previous hospitalstay > 2 days | 1.4 [0.1–2.2] | 0.080 | 2.0 [1.2–3.3] | **0.006** |
| **Ward/Unit** | | 0.000 | | **0.003** |
| • Postoperative care ward/Dafra | 1 | | 1 | |
| • Postoperative care ward/Do | 0.6 [0.3–1.4] | | 0.7 [0.2–2.1] | |
| • Gyn & obst postoperative care ward/CHUSS | 0.8 [0.4–1.6] | | 0.6 [0.3–1.3] | |
| • Neonatal ward/CHUSS | 3.4 [1.8–6.4] | | 6.6 [1.1–40.2] | |
| • ICU/CHUSS | 3.8 [1.9–7.9] | | 3.3 [1.3–8.5] | |

Gyn & obst = Gynecology & obstetric; ICU = Intensive care unit, p-value* = global p-value; cOR = crude odd ration;aOR = adjusted odd ratio; CHUSS=Sourô Sanou Teaching Hospital

## HAI Outcomes

The mean follow-up duration of included patients was 9.4 days (SD: 9.2) with a range of 2 to 30 days. It was twice as long for HAIs (15.7 days vs 7.3 days; p = 0.000). The mean length of hospital stay was 7.8 days (SD: 7.9).It was longer for HAI cases (8.9 days vs 7.4 days; p = 0.039). Of the 664 patients included, we observed 569 recoveries (85.7%), 6 discharges against medical advice (0.9%), 62 deaths (9.3%), and 27 lost to follow-up (4.1%). Deaths were observed only in the neonatal ward (n = 42) and ICU (n = 20). The incidence of death was higher in patients with HAI (18.1%; 95%CI:12.1–24.4) than in patients without HAI (6.4%;95%CI: 4.3–8.5) p = 0.000.HAI mortality was 45.4%(95%CI:27.5–63.4) in the ICU and 21.4% HAI mortality (95%CI: 11.6–31.3) in the neonatal ward (p = 0.019) (S4 Table)

## Discussion

Our study reported a cumulative incidence rate of HAIs of 25% (95%CI: 21.7%–28.3%) or an incidence density rate of 36.7 (95%CI:31.7–42.9) per 1000 patient-days.The highest incidences of HAIs were observed in the ICU (45.2%; 95%CI: 33.5–56.9%) and in the neonatal ward (42.2%; 95%CI: 34.6–49.7%). Factors associated with HAIs were previous hospital stay longer than 2 days, male sex, admission to the neonatal wardor ICU. HAIs were associated with increased length of follow-up, hospitalization, and mortality.

The incidence of HAIs in our study was comparable to the 28.15 [95% CI: 24.40–32.30] per 1000 patient days reported by Ali et al [17] in a longitudinal study of 1015 patients at Jimma University Medical in Ethiopia. It was also consistent with the cumulative incidence of 5.7% to 45.8% found in Africa by Nedja et al [18]. However, this incidence was higher than the 9.7 (95% CI: 7.1–12.9) per 1000 patient days reported by Chernet et al [19] in a longitudinal study in Ethiopia. The incidence of HCAI in our study was also much higher than that reported by Abubakar et al [9], Mpinda-Joseph et al [10], Some [15] and Olivier et al [20], who reported prevalences of 12.76%, 13.45%, 10% and 9.9% respectively.Variations in the incidence/prevalence of HAIs between studies could be explained by methodological differences, such as the type of study, the choice of infections studied and the definition of HAI cases. It is recognized that longitudinal or incidence studies, such as the present one, offer greater precision in assessing the extent of these infections [13].But these studies are time-consuming and require more resources than the prevalence studies that are more commonly conducted. Similarly, the level of development of healthcare infrastructures and the presence or absence of infection prevention and control (IPC) program in the healthcare structure are factors that can determine the extent of HAIs. For example, the incidence of HAIs reported in developed countries (3.5% to 12%) was lower than in developing countries [2,21]. As in our study, Alemu et al [11] and Ali et al [17] found the highest prevalence of HAIs in neonatal and intensive care units. The high frequency of invasive care in these units, combined with the immaturity or dysfunction of the immune system of the patients admitted, makes them more susceptible to HAIs.

The performance of microbiological culture in our study (57%) was consistent with those reported in the literature (41–86%) [22] but remains much higher than that reported by Ezzi et al. in Tunisia (29.6%) [23]. The poor performance of culture in our study could be explained by the common practice in the neonatal ward of initiating systematic empirical antibiotic therapy, prior to obtaining bacterial cultures. However, the predominance of Enterobacterales in HAIs has been widely reported in the literature [5,6,24–26] and was confirmed in our study. In addition, this study highlighted a relatively high prevalence of extended-spectrum beta-lactamase producing Enterobacterales (ESBLE) 38.9%, which is consistent with data from the literature [6,27]. These findings suggest the need to rationalize the use of antibiotics in our hospitals by establishing therapeutic protocols based on antimicrobial resistance surveillance data.

The risk factors for HAI in our study are consistent with the results of numerous studies showing an association between length previous hospital stay,male sex, neonatal or ICU admission and the occurrence of these infections [9,17,28]. Long hospital stays mean prolonged exposure to pathogens in the hospital environment, as well as an increased likelihood of invasive medical procedures, resulting in a higher risk of HAI. The association of male sex with an increased risk of HAIs is a controversial topic. However, differences between men and women in terms of personal hygiene practices, the frequency of comorbidities, and anatomy may explain this association [29–31]. Patients admitted to neonatal,and ICU are often vulnerable due to their compromised immune system. In addition, the high number of critically ill patients and the complex medical procedures performed in these units make them a breeding ground for pathogens.

Our study, like many others, shows an association between HAIs and increased length of hospital stay, comorbidity and high mortality [2]. Our overall case fatality rate of HAIswas higher than the 14% reported by Gidey et al [32] in Ethiopia.The inclusion of high-mortality wards such as ICU and neonatology in our study may explain the superior mortality in our study. However, our case fatality rate of HAIsin ICU is similar to the case fatality rate ofMerzougui et al (44.7%) [24] in ICU in Tunisia and thatof Rimaz et al in the Islamic Republic of Iran (45.2%) [26]. Data on HAI outcomes vary according to study methodology, accuracy of HAI case identification strategy, the typology of HAIs studied, and the quality of care. Prospective, longitudinal studies such as ours are more appropriate than retrospective studies for assessing HAI outcomes.

The main limitation of this study was its relatively short inclusion period (two to four months), which does not allow analysis of seasonal variations in the incidence of HAIs. In addition, the present study does not consider environmental and behavioral factors of healthcare workers in the analysis of risk factors of HAIs. However, its prospective longitudinal design has the advantage of allowing the analysis of the risk factors of HAIs.

## Conclusion

The incidence of HAIs was relatively high in the three hospitals in Bobo Dioulasso, Burkina Faso. The pathogens identified in these infections were mainly Enterobacterales, with 38.9%being ESBL. Risk factors for HAI included length of previous hospital stay, male sex, neonatal or ICU admission. These findings underscore the importance of strengthening infection prevention and control measures in hospitals, particularly in high-risk wards. It is essentialto implement rigorous protocols for patient management, as well as continuous training programs for medical and paramedical staff. Additionally, enhanced surveillance of HAI and judicious use of antibiotics are essential to reduce the spread of resistant pathogens in general and Enterobacterales in particular in Bobo-Dioulasso hospital.

## Supporting information

**S1 Appendice. Microbial identification and antimicrobial susceptibility, testing, multicenter longitudinal survey, Bobo-Dioulasso, 2022.**
(DOCX)

**S2 Table. Definitions of HAI Cases Monitored, multicenter longitudinal survey, Bobo-Dioulasso, 2022.**
(DOCX)

**S3 File. Data base.**
(XLS)

**S4 Table. Distribution of adult and pediatric patients according to HAI data, multicenter longitudinal survey, Bobo-Dioulasso 2022.**
(DOCX)

## Acknowledgments

We would like to thank the data collectors, the staff of Sourô Sanou Teaching Hospital, Do and Dafra District Hospitals. Our gratitude extends to the patients for their participation in this study.

## Author contributions

**Conceptualization:** Arsene Hema, Satouro Arsène Somé, Odilon Kaboré, Soufiane Sanou, Léon Savadogo.

**Data curation:** Arsene Hema, Odilon Kaboré.

**Formal analysis:** Arsene Hema, Satouro Arsène Somé, Armel Poda, Ziemlé Clément Meda.

**Investigation:** Arsene Hema, Satouro Arsène Somé, Odilon Kaboré.

**Methodology:** Arsene Hema.

**Supervision:** Soufiane Sanou, Ziemlé Clément Meda, Abdoul Salam Ouedraogo, Léon Savadogo.

**Validation:** Satouro Arsène Somé, Armel Poda, Léon Savadogo.

**Writing – original draft:** Arsene Hema, Satouro Arsène Somé, Odilon Kaboré, Soufiane Sanou.

**Writing – review & editing:** Armel Poda, Ziemlé Clément Meda, Abdoul Salam Ouedraogo, Léon Savadogo.

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
