## [Decision Letter · Decision Letter 0]

9 Oct 2024

PONE-D-24-26487Risk and outcomes of healthcare-associated infections in three hospitals in Bobo Dioulasso, 2022 (Burkina Faso): a longitudinal studyPLOS ONE

Dear Dr. HEMA,

Thank you for submitting your manuscript to PLOS ONE. After careful consideration, we feel that it has merit but does not fully meet PLOS ONE’s publication criteria as it currently stands. Therefore, we invite you to submit a revised version of the manuscript that addresses the points raised during the review process.

**Please carefully revise your manuscript according to the reviewers’ comments below. Please also consider the following comment: The percentage of preventable HAI mentioned in the abstract and introduction as well as the references used to support this is debatable. Please delete this statement from the abstract. Please revise the statement in the introduction and considere the following aspects: A publication form Germany https://www.sciencedirect.com/science/article/pii/S0720337310001130?via%3Dihub suggests that only 30% of HAI are being preventable by hygiene measures while an US-American study suggests different proportions for different HAI types (65 – 70% for CABSI / CAUTI, 55% for VAP and SSI, https://pubmed.ncbi.nlm.nih.gov/21460463/ .   Please re-check the references you used.** **Line 50: Please correct "ESBL" producer. **

We look forward to receiving your revised manuscript.

Kind regards,

Luisa Anna Denkel

Academic Editor

PLOS ONE

**Journal Requirements:**

3. In the online submission form, you indicated that The datasets, including deidentified provider quantitative and qualitative data, used and/or analyzed during the current study are available from the corresponding author upon reasonable request.

Reviewers' comments:

Reviewer's Responses to Questions

**Comments to the Author**

1. Is the manuscript technically sound, and do the data support the conclusions?

Reviewer #1: Yes

Reviewer #2: Partly

2. Has the statistical analysis been performed appropriately and rigorously? 

Reviewer #1: I Don't Know

Reviewer #2: N/A

3. Have the authors made all data underlying the findings in their manuscript fully available?

Reviewer #1: Yes

Reviewer #2: No

4. Is the manuscript presented in an intelligible fashion and written in standard English?

Reviewer #1: Yes

Reviewer #2: No

5. Review Comments to the Author

**Reviewer #1:**  The Title: The use of parentheses in this title is inconvenient.

Ligne 36 et 92, 131, 136 : Correct writting of dates (superscripts). Also review the entire text.

Line 37: Define the abbreviation DO or write it as the name (Do). Harmonize writing throughout the text.

Line 87: Risk factors (in the title) or etiological factors (as aim): For me, It seems that the terms can be nuanced.

Line 185: Sentence to be reviewed.

Lines 199 and 350 (ethical aspects): the reference number of the national ethics committee must be mentioned.

If you say: "We obtained informed consent, written and signed from parents or guardians for minors", please also specify if the consent of minors has also been obtained.

Table 3: cOR and aOr must be defined in the legend.

Line 244 (point 3.3): Why were the proportions estimated based on 45 microorganisms instead of the 47 identified?

Lines 243 and 306: How was the selection of HAIS for bacterial identification made (79 out of 166)?

Enterobacteriaceae, Enterobacteria: Review use in context.

Lines 292 -305: A clearer explanation of the incidence found can be made as you have just made comparisons but without explaining why the difference in results observed. Is the comparison made with prevalence or incidence studies?

Moreover, for the articles with which the comparison is made, it would be preferable to put the results found instead of just quoting the articles in question (lines 292 -305 and 306-315).

Ligne 303“The lack of infection control policies and trained personnel and the overcrowding of our hospitals may explain the high incidence of HAIs in our study” This aspect was not mentioned in the results section.

Line 326-334 (penultimate paragraph of the discussion): You have mentioned the first and last names of the authors as references, which should not be the case. Please refer to the authors' guide for references in the body of the text.

In conclusion: You conclude that the incidence is high. Are there any acceptable thresholds?

Why separate enterobacteria from gram-negative bacilli in Figure 2?

**Reviewer #2:**  General comments

Thank you so much for the opportunity to review this manuscript. It presents an interesting and important study on healthcare-associated infections (HAIs) in three hospitals in Burkina Faso, a region in West Africa that faces significant healthcare challenges, including HAIs. While the study addresses a critical need for data from resource-constrained settings, several areas require substantial revisions to improve the quality and clarity of the research presentation.

Major comments

1. Language and grammar

i. It would be beneficial to have the entire manuscript reviewed by a native English speaker or a professional editor, as there are numerous grammatical errors, spelling mistakes, inconsistent use of abbreviations and unnecessary use of bullet points (e.g., "May 1th" should be "May 1st") that distract the reader.

2. Abstract

i. In the "Results" section, clarify the statistical significance (e.g., p-values) of the associations mentioned.

3. Materials and Methods

i. Study design: Consider changing the "Study design and location" subtitle to simply "Study design" and ‘‘setting’’. Clearly outline the longitudinal, multicentre nature of the study upfront.

ii. Sampling: I suggest that the authors should include a detailed description of the sampling methods. Currently, the focus is on the wards surveyed without explaining the rationale for choosing these wards – this need to be addressed

iii. Use "exclusion criteria" instead of "non-inclusion criteria." Additionally, avoid using bullet points when describing inclusion and exclusion criteria, as this can distract the reader. These sections should be revised.

iv. Surveillance and data collection: Define the operational definitions from the French Technical Committee more clearly. What kind of modifications were made to the operational definitions? This need to be explained.

v. Provide more clarity on the method used for identifying cases and how the study ensured the reliability of data collection.

vi. Data analysis: Specify the software versions used for statistical analysis. Clarify the process by which the significance threshold was set at p < 0.05.

vii. HAI classification: The manuscript mentions targeting specific infections like SSIs and neonatal infections. However, there is a need to provide more detail about how these infections were classified and confirmed.

viii. Outcomes: The discussion around HAI outcomes, such as length of stay, needs to address the potential biases and limitations in data collection and interpretation.

4. Results

i. Presentation: Replace terms like "third of the sample" with specific numbers and percentages to ensure clarity.

ii. Analysis of subgroups: Provide a more detailed breakdown of HAI data by age group, sex, and hospital ward. Consider reorganising the data to separately present outcomes for children and adults.

iii. Incidence rates: The text should clarify whether the reported incidence density rates are crude or adjusted. Additionally, describe how these rates compare to those found in other regional studies to add context.

5. Discussion

i. The discussion should be rewritten using a standard scientific format, including a comparison with previous research and an exploration of potential reasons for the differences observed in this study.

ii. Avoid repetition of results in the discussion section.

iii. The recommendations section seems overly general. It would benefit from being directly linked to the study's results to provide more actionable insights for healthcare facilities in similar settings.

6. Ethics statement

i. Expand on the ethics approval process, including the measures taken to protect patient confidentiality and the consent procedure for different age groups (especially neonates).

7. Conclusions

i. While the conclusions section appropriately summarises the key findings, it should be slightly expanded to include more specific recommendations for policy and practice based on the study’s results.

Specific comments

i. Page 4, Line 66: Consider revising to "…HAIs are caused by bacteria, viruses, or fungi that develop resistance..." and the statement should be supported with a reference.

ii. Page 9, Line 162-163: Provide more detail on how operational definitions were modified for this study. Modified operational definitions can be provided as an appendix.

iii. Clarify how patient residence (urban/rural) might influence HAI risk.

Recommendations

i. Restructure the methods section: Clearly define study design, setting, data collection, and analysis to facilitate reproducibility.

ii. Data analysis: Expand the data analysis section to include more in-depth explanations of the statistical methods used.

iii. Supplementary materials: Include more detailed information in the supplementary materials if some content (e.g., specific diagnostic criteria) makes the main text too dense.

Best wishes!

6. PLOS authors have the option to publish the peer review history of their article (what does this mean? ). If published, this will include your full peer review and any attached files.

**Do you want your identity to be public for this peer review?** For information about this choice, including consent withdrawal, please see our Privacy Policy .

Reviewer #1: **Yes: ** Dr Tani SAGNA

Reviewer #2: No

---

## [Author Response · Author response to Decision Letter 0]

5 Nov 2024

Response to reviewers

Dear Editor and Reviewers,

Thank you for agreeing to review this manuscript, and your comments and observations are pertinent. In this letter, we would like to provide you with some clarifications and responses to your questions.

Academic editor

The percentage of preventable HAI mentioned in the abstract and introduction as well as the references used to support this is debatable. Please delete this statement from the abstract. Please revise the statement in the introduction and considere the following aspects: A publication form Germany https://www.sciencedirect.com/science/article/pii/S0720337310001130?via%3Dihub suggests that only 30% of HAI are being preventable by hygiene measures while

Response: We have replaced the frequency previously mentioned in the abstract and in the text by the one reported by the German study, as recommended.

Line 50: Please correct "ESBL" producer.

Response: Correction applied

Your ethics statement should only appear in the Methods section of your manuscript. If your ethics statement is written in any section besides the Methods, please delete it from any other section.

Response: Our ethics statement is in Methods section.

Reviewer #1:

The Title: The use of parentheses in this title is inconvenient. Ligne 36 et 92, 131, 136 : Correct writing of dates (superscripts). Also review the entire text

Response: Parentheses were removed and dates were corrected as well.

Line 37: Define the abbreviation DO or write it as the name (Do). Harmonize writing throughout the text.

Response: We corrected and harmonized the writing of the name of the District Hospital using ''Do''.

Line 87: Risk factors (in the title) or etiological factors (as aim): For me, It seems that the terms can be nuanced.

Response: Correction was made: etiological factor was replaced by risk factor

Line 185: Sentence to be reviewed.

Response: Sentence revised

Lines 199 and 350 (ethical aspects): the reference number of the national ethics committee must be mentioned.

Response: ethics committee approval reference number was added

If you say: "We obtained informed consent, written and signed from parents or guardians for minors", please also specify if the consent of minors has also been obtained.

Response: We obtained written and signed informed consent from parents or guardians for newborns and minors. In addition to parental consent, we have obtained the assent of children over the age of 12.

Table 3: cOR and aOr must be defined in the legend.

Response: Correction made: cOR= crude odd ratio; aOR : adjusted odd ratio

Line 244 (point 3.3): Why were the proportions estimated based on 45 microorganisms instead of the 47 identified?

Response: This is an error, 45 microorganisms were identified. The correction was made.

Lines 243 and 306: How was the selection of HAIS for bacterial identification made (79 out of 166)?

Response: Not all HAI cases underwent biologic testing. We have randomly selected 79 patients’ sample for biological testing.

Enterobacteriaceae, Enterobacteria: Review use in context.

Response: Correction made, replace “Enterobacteria” with “Enterobacteriaceae

Lines 292 -305: A clearer explanation of the incidence found can be made as you have just made comparisons but without explaining why the difference in results observed. Is the comparison made with prevalence or incidence studies?

Moreover, for the articles with which the comparison is made, it would be preferable to put the results found instead of just quoting the articles in question (lines 292 -305 and 306-315).

Response: We have expanded the discussion to address your comments.

Ligne 303“The lack of infection control policies and trained personnel and the overcrowding of our hospitals may explain the high incidence of HAIs in our study” This aspect was not mentioned in the results section.

Response: correction made. The statement was deleted

Line 326-334 (penultimate paragraph of the discussion): You have mentioned the first and last names of the authors as references, which should not be the case. Please refer to the authors' guide for references in the body of the text.

Response: correction made in line with the authors’ guide for references in the body of the text

In conclusion: You conclude that the incidence is high. Are there any acceptable thresholds?

Response: Basically, there is no threshold. However, we believe that for quality care, the frequency of HAIs should be closer to 5%.

Why separate enterobacteria from gram-negative bacilli in Figure 2?

Response: Enterobacteriaceae are indeed Gram-negative, but this family of bacteria is of particular interest in antimicrobial resistance monitoring because of the rapid growth of ESBLE.

Reviewer #2: General comments

Thank you so much for the opportunity to review this manuscript. It presents an interesting and important study on healthcare-associated infections (HAIs) in three hospitals in Burkina Faso, a region in West Africa that faces significant healthcare challenges, including HAIs. While the study addresses a critical need for data from resource-constrained settings, several areas require substantial revisions to improve the quality and clarity of the research presentation.

Major comments

1. Language and grammar

i. It would be beneficial to have the entire manuscript reviewed by a native English speaker or a professional editor, as there are numerous grammatical errors, spelling mistakes, inconsistent use of abbreviations and unnecessary use of bullet points (e.g., "May 1th" should be "May 1st") that distract the reader.

Response: The manuscript has been proofread and corrections was made to make the text more digestible.

2. Abstract

i. In the "Results" section, clarify the statistical significance (e.g., p-values) of the associations mentioned.

Response: We have presented the adjusted odd ratios with their 95% confidence intervals. Normally, these are sufficient to show the strength of the association between HAI and patient characteristics. If a 95% CI does not contain 1, then the association is significant (p<5%).

3. Materials and Methods

i. Study design: Consider hanging the "Study design and location" subtitle to simply "Study design" and ‘‘setting’’. Clearly outline the longitudinal, multicentre nature of the study upfront.

Response: correction made accordingly

ii. Sampling: I suggest that the authors should include a detailed description of the sampling methods. Currently, the focus is on the wards surveyed without explaining the rationale for choosing these wards – this need to be addressed

Response: In this manuscript we present surveillance data, so we did not sample. We included all eligible cases and then measured the duration of surveillance.

iii. Use "exclusion criteria" instead of "non-inclusion criteria." Additionally, avoid using bullet points when describing inclusion and exclusion criteria, as this can distract the reader. These sections should be revised.

Response: Revisions was made

iv. Surveillance and data collection: Define the operational definitions from the French Technical Committee more clearly. What kind of modifications were made to the operational definitions? This need to be explained.

Response: We've included the operational definitions in the supplementary materials to avoid overloading the article. Please refer to the supplementary materials

v. Provide more clarity on the method used for identifying cases and how the study ensured the reliability of data collection.

Response: As described in the methodology, data collection teams visited the selected wards three times a week to review patient records and registers and to talk to staff and patients.

HAI cases identified by the investigators based on operational definitions were validated by a team consisting of the principal investigator, the attending physician, and the investigator teams.

vi. Data analysis: Specify the software versions used for statistical analysis. Clarify the process by which the significance threshold was set at p < 0.05.

Response: correction made

vii. HAI classification: The manuscript mentions targeting specific infections like SSIs and neonatal infections. However, there is a need to provide more detail about how these infections were classified and confirmed.

Response: Patients were classified on the basis of clinical data that was validated by the teams which comprise the treating physician and the investigative team members. They used case definitions and, where appropriate, biological data.

viii. Outcomes: The discussion around HAI outcomes, such as length of stay, needs to address the potential biases and limitations in data collection and interpretation.

Response: We have presented these parameters as limitations of this study in the discussion.

4. Results

i. Presentation: Replace terms like "third of the sample" with specific numbers and percentages to ensure clarity.

Response: correction made

ii. Analysis of subgroups: Provide a more detailed breakdown of HAI data by age group, sex, and hospital ward. Consider reorganising the data to separately present outcomes for children and adults.

Response: The sub-group analysis is shown in Table 1. It shows the distribution of HAIs cases by sex and age and ward. We have provided a table describing the distribution of adult and pediatric patients according to HAI data as supplementary information.

iii. Incidence rates: The text should clarify whether the reported incidence density rates are crude or adjusted. Additionally, describe how these rates compare to those found in other regional studies to add context.

Response: these are crude incidence rates.

5. Discussion

i. The discussion should be rewritten using a standard scientific format, including a comparison with previous research and an exploration of potential reasons for the differences observed in this study.

Response: The discussion was rewritten accordingly

ii. Avoid repetition of results in the discussion section.

Response: Comment addressed

iii. The recommendations section seems overly general. It would benefit from being directly linked to the study's results to provide more actionable insights for healthcare facilities in similar settings.

Response: correction made, more specific recommendations were formulated

6. Ethics statement

i. Expand on the ethics approval process, including the measures taken to protect patient confidentiality and the consent procedure for different age groups (especially neonates).

Response: correction made

7. Conclusions

i. While the conclusions section appropriately summarises the key findings, it should be slightly expanded to include more specific recommendations for policy and practice based on the study’s results.

Response: specific recommendations were included in the conclusion

Specific comments

i. Page 4, Line 66: Consider revising to "…HAIs are caused by bacteria, viruses, or fungi that develop resistance..." and the statement should be supported with a reference.

Response: reference provided

ii. Page 9, Line 162-163: Provide more detail on how operational definitions were modified for this study. Modified operational definitions can be provided as an appendix.

Response: correction made

iii. Clarify how patient residence (urban/rural) might influence HAI risk.

Response: Residence (urban/rural) had no effect on the risk of HAI in this study. However, some studies show that rural residence is associated with a higher risk of HAI. The explanation for this association remains unclear, but some authors believe that the lack of sanitary infrastructure in rural areas and the delay in access to care, especially for pregnant women, could be the cause.

General comments

Thank you so much for the opportunity to review this manuscript. It presents an interesting and important study on healthcare-associated infections (HAIs) in three hospitals in Burkina Faso, a region in West Africa that faces significant healthcare challenges, including HAIs. While the study addresses a critical need for data from resource-constrained settings, several areas require substantial revisions to improve the quality and clarity of the research presentation.

Major comments

1. Language and grammar

i. It would be beneficial to have the entire manuscript reviewed by a native English speaker or a professional editor, as there are numerous grammatical errors, spelling mistakes, inconsistent use of abbreviations and unnecessary use of bullet points (e.g., "May 1th" should be "May 1st") that distract the reader.

Response: Language issues have been addressed by proofreading the manuscript.

2. Abstract

i. In the "Results" section, clarify the statistical significance (e.g., p-values) of the associations mentioned.

Response: The p-value, or statistical significance threshold, is the probability that the observed results are due to the hazard. Conventionally, this threshold is set at 5%.

3. Materials and Methods

i. Study design: Consider changing the "Study design and location" subtitle to simply "Study design" and ‘‘setting’’. Clearly outline the longitudinal, multicentre nature of the study upfront.

Response: correction made

ii. Sampling: I suggest that the authors should include a detailed description of the sampling methods. Currently, the focus is on the wards surveyed without explaining the rationale for choosing these wards – this need to be addressed

Response: In this manuscript we present surveillance data, so we did not sample. We included all eligible cases and then measured the duration of surveillance.

iii. Use "exclusion criteria" instead of "non-inclusion criteria." Additionally, avoid using bullet points when describing inclusion and exclusion criteria, as this can distract the reader. These sections should be revised.

Response: correction made

iv. Surveillance and data collection: Define the operational definitions from the French Technical Committee more clearly. What kind of modifications were made to the operational definitions? This need to be explained.

Response: We didn't want to include these definitions in the article so as not to overload it. However, we have included them as supplementary material. As a modification, we did not include the different categories of site infection (superficial, deep, organ).

v. Provide more clarity on the method used for identifying cases and how the study ensured the reliability of data collection.

Response: As described in the methodology, data collection teams visited the selected wards three times a week to review patient records and registers and to talk to staff and patients. Selection of HAI cases was based on a flow chart (see Supplementary materials). HAI cases were validated by the survey team and the attending physician.

vi. Data analysis: Specify the software versions used for statistical analysis. Clarify the process by which the significance threshold was set at p < 0.05.

Response: Correction made. stata version 13 were used for data analysis. P-value statistical significance level is the probability that the observed results are due to hazard. It is conventionally set at 5%.

vii. HAI classification: The manuscript mentions targeting specific infections like SSIs and neonatal infections. However, there is a need to provide more detail about how these infections were classified and confirmed.

Response: classification of HAIs is based on case definitions and was validated by investigators and treating physicians.

viii. Outcomes: The discussion around HAI outcomes, such as length of stay, needs to address the potential biases and limitations in data collection and interpretation.

Response: We have presented these parameters as limitations of this study in the discussion.

4. Results

i. Presentation: Replace terms like "third of the sample" with specific numbers and percentages to ensure clarity.

Response: correction made

ii. Analysis of subgroups: Provide a more detailed breakdown of HAI data by age group, sex, and hospital ward. Consider reorganising the data to separately present outcomes for children and adults.

Response: The sub-group analysis is shown in Table 1. It shows the distribution of HAIs

---

## [Decision Letter · Decision Letter 1]

26 Nov 2024

PONE-D-24-26487R1Risk and outcomes of healthcare-associated infections in three hospitals in Bobo Dioulasso, Burkina Faso, 2022: a longitudinal studyPLOS ONE

Dear Dr. HEMA,

Thank you for re-submitting your manuscript to PLOS ONE. After careful consideration, we feel that it has merit but does not fully meet PLOS ONE’s publication criteria as it currently stands. Therefore, we invite you to submit a revised version of the manuscript that addresses the points raised during the review process.

Please consider the final suggestions made by reviewer 2 to further improve the quality of your work. 

We look forward to receiving your revised manuscript.

Kind regards,

Luisa Anna Denkel

Academic Editor

PLOS ONE

Journal Requirements:

Reviewers' comments:

Reviewer's Responses to Questions

**Comments to the Author**

1. If the authors have adequately addressed your comments raised in a previous round of review and you feel that this manuscript is now acceptable for publication, you may indicate that here to bypass the “Comments to the Author” section, enter your conflict of interest statement in the “Confidential to Editor” section, and submit your "Accept" recommendation.

Reviewer #1: All comments have been addressed

Reviewer #2: (No Response)

2. Is the manuscript technically sound, and do the data support the conclusions?

Reviewer #1: Yes

Reviewer #2: Yes

3. Has the statistical analysis been performed appropriately and rigorously? 

Reviewer #1: I Don't Know

Reviewer #2: Yes

4. Have the authors made all data underlying the findings in their manuscript fully available?

Reviewer #1: Yes

Reviewer #2: Yes

5. Is the manuscript presented in an intelligible fashion and written in standard English?

Reviewer #1: Yes

Reviewer #2: No

6. Review Comments to the Author

Reviewer #1: (No Response)

Reviewer #2: I would like to commend the authors for their efforts in addressing most of the issues raised earlier, which is highly commendable. However, I believe there is still room for improvement, particularly in refining the language to enhance clarity and flow. Below are some minor issues that could serve as a guide to address areas that have not been adequately resolved, which will improve the manuscript.

1. Writing numbers in scientific writing

In scientific writing, numbers from one to nine are typically written out in words (e.g., one, two, three) rather than as numerals (e.g., 1, 2, 3). Numbers 10 and above should be written as numerals (e.g., 10, 11, 20, etc.). Therefore, the notation such as ‘four (04)’ is redundant and should be simplified to just "four." This should be corrected throughout the manuscript.

2. Lines 124-126

The sentences ‘In 2022, the gynecology and obstetrics postoperative care ward of Sourô Sanou Teaching Hospital received 1,946 women, including 1,433 cesareans. This unit has 30 inpatient beds. The staff consisted of 16 physicians, four (04) nurses and 21 midwives.’ This can be improved as follows:

• As earlier noted, numbers between 1 and 9 should be written out in full, so "four (04) nurses" should simply be "four nurses" to avoid unnecessary distractions.

• The phrase "including 1,433 cesareans" is unclear. It might imply that all 1,946 women had cesareans, but the word "including" suggests a subset. To improve clarity, I would rephrase it to something like:

‘In 2022, the gynecology and obstetrics postoperative care ward of Sourô Sanou Teaching Hospital treated 1,946 women, 1,433 of whom had cesarean sections. The unit has 30 inpatient beds and a staff of 16 physicians, four nurses, and 21 midwives.’

Note: this is just an example

3. Lines: 152-156

The sentence ‘Information collected from these sources was supplemented by information from patients and nursing staff and recorded on individual data collection sheets which included, among others: socio demographic data (age, sex, origin, profession), factors risk related to care, initial or curative antibiotic therapy, HAI type, pathogen in question, development.’

• The phrase "factors risk related to care" is grammatically incorrect. It should be revised to something like "risk factors related to care."

• The sentence is quite long and could benefit from breaking it into two sentences to improve readability

• The phrase "included, among others" could be revised for conciseness to just "included." The reader will infer that the list is not exhaustive.

• The word "development" is vague. Specify whether it refers to patient outcomes, the progression of an infection, or something else.

4. In line 261-265

• Like other bacteria listed out the Escherichia coli should be italicized

• Use "spp." for species abbreviation and correct "spp1" to spp (Pantoea spp1)

• Use ‘spp’ instead of ‘sp’ (Enterococcus sp)

5. Discussion: The discussion section still contains repeated results, particularly in the first and second paragraphs, which is unnecessary. The discussion should focus on interpreting and analyzing your findings rather than restating them. While it is acceptable to highlight key results, summarizing the entire results in this section is not ideal. Please revise to ensure the discussion provides meaningful insights and context for your results.

6. Line 328: The abbreviation ‘AIS’ is not defined anywhere in the manuscript, if it refers to HAIs kindly correct it.

7. Lines 344-346: ‘Systematic empirical antibiotic therapy, whether prophylactic or curative, sometimes prior to bacterial culture, a common practice in the neonatal ward, could explain the poor performance of culture in our study.’

It is important to note that requesting a bacterial culture before surgical prophylaxis is unnecessary. Antibiotic use can generally be categorized into three main indications: empiric, targeted/definitive, and prophylactic. Prophylactic antibiotics are typically guided by the type of surgical wound (clean-contaminated or contaminated) and the likely pathogens expected during the procedure. There is ongoing debate about the necessity of prophylactic antibiotics in clean wounds, while for contaminated wounds, prophylaxis is not indicated because these wounds require treatment for an established infection. As such, obtaining a bacterial culture is not a prerequisite for surgical prophylaxis, which is specifically intended to prevent infections from anticipated pathogens rather than treat an existing one. I would suggest you revise the sentence to something like:

‘The poor performance of culture in our study could be explained by the common practice in the neonatal ward of initiating systematic empirical antibiotic therapy, prior to obtaining bacterial cultures.’

Note: This is just an example

8. Lines 358-359: ‘However, differences between men and women in terms of personal hygiene practices, of comorbidities frequency and anatomy may explain this association.’

• The phrase "in terms of personal hygiene practices, of comorbidities frequency and anatomy" is cumbersome and could be streamlined for better clarity.

• The phrase "comorbidities frequency" should be "the frequency of comorbidities" for proper grammar

I would revise the sentence as ‘However, differences between men and women in terms of personal hygiene practices, the frequency of comorbidities, and anatomy may explain this association.’

Note: This is just an example

Best of luck

7. PLOS authors have the option to publish the peer review history of their article (what does this mean? ). If published, this will include your full peer review and any attached files.

**Do you want your identity to be public for this peer review?** For information about this choice, including consent withdrawal, please see our Privacy Policy .

Reviewer #1: No

Reviewer #2: **Yes: ** Muhammad Augie Bashar

---

## [Author Response · Author response to Decision Letter 1]

29 Nov 2024

Response to reviewers

Dear Editor and Reviewers,

Thank you for agreeing to review this manuscript, and your comments and observations. Thank you for your suggestions and recommendations. In this letter, we would like to provide you with some clarifications and responses to your questions.

I would like to commend the authors for their efforts in addressing most of the issues raised earlier, which is highly commendable. However, I believe there is still room for improvement, particularly in refining the language to enhance clarity and flow. Below are some minor issues that could serve as a guide to address areas that have not been adequately resolved, which will improve the manuscript.

Recommendation 1: Writing numbers in scientific writing

In scientific writing, numbers from one to nine are typically written out in words (e.g., one, two, three) rather than as numerals (e.g., 1, 2, 3). Numbers 10 and above should be written as numerals (e.g., 10, 11, 20, etc.). Therefore, the notation such as ‘four (04)’ is redundant and should be simplified to just "four." This should be corrected throughout the manuscript.

Response: We've written all the numbers below 10 in words.

Recommendation 2: Lines 124-126

The sentences ‘In 2022, the gynecology and obstetrics postoperative care ward of Sourô Sanou Teaching Hospital received 1,946 women, including 1,433 cesareans. This unit has 30 inpatient beds. The staff consisted of 16 physicians, four (04) nurses and 21 midwives.’ This can be improved as follows:

• As earlier noted, numbers between 1 and 9 should be written out in full, so "four (04) nurses" should simply be "four nurses" to avoid unnecessary distractions.

• The phrase "including 1,433 cesareans" is unclear. It might imply that all 1,946 women had cesareans, but the word "including" suggests a subset. To improve clarity, I would rephrase it to something like:

‘In 2022, the gynecology and obstetrics postoperative care ward of Sourô Sanou Teaching Hospital treated 1,946 women, 1,433 of whom had cesarean sections. The unit has 30 inpatient beds and a staff of 16 physicians, four nurses, and 21 midwives.’

Note: this is just an example

Response: This recommendation has been taken into account. The sentences have been modified as proposed.

Recommendation 3 : Lines: 152-156

The sentence ‘Information collected from these sources was supplemented by information from patients and nursing staff and recorded on individual data collection sheets which included, among others: socio demographic data (age, sex, origin, profession), factors risk related to care, initial or curative antibiotic therapy, HAI type, pathogen in question, development.’

• The phrase "factors risk related to care" is grammatically incorrect. It should be revised to something like "risk factors related to care."

• The sentence is quite long and could benefit from breaking it into two sentences to improve readability

• The phrase "included, among others" could be revised for conciseness to just "included." The reader will infer that the list is not exhaustive.

• The word "development" is vague. Specify whether it refers to patient outcomes, the progression of an infection, or something else.

Response:

• The term " factors risk related to care" has been replaced by "risk factors related to care".

• The long sentence has been reformulated into two sentences.

• The word "including" has been replaced by "including".

• Replace the word "development" with "patient outcomes".

Recommendation 4 In line 261-265

• Like other bacteria listed out the Escherichia coli should be italicized

• Use "spp." for species abbreviation and correct "spp1" to spp (Pantoea spp1)

• Use ‘spp’ instead of ‘sp’ (Enterococcus sp)

Response:

• Escherichia coli has been italicized

• We have used “spp” for Pantoea et Enterococcus

Recommendation 5: Discussion: The discussion section still contains repeated results, particularly in the first and second paragraphs, which is unnecessary. The discussion should focus on interpreting and analyzing your findings rather than restating them. While it is acceptable to highlight key results, summarizing the entire results in this section is not ideal. Please revise to ensure the discussion provides meaningful insights and context for your results.

Response: We have summarized the key findings at the beginning of the discussion.

Recommendation 6 : Line 328: The abbreviation ‘AIS’ is not defined anywhere in the manuscript, if it refers to HAIs kindly correct it.

Response: Correction “HAIs” instead of “AIS”

Recommendation 7: Lines 344-346: ‘Systematic empirical antibiotic therapy, whether prophylactic or curative, sometimes prior to bacterial culture, a common practice in the neonatal ward, could explain the poor performance of culture in our study.’

It is important to note that requesting a bacterial culture before surgical prophylaxis is unnecessary. Antibiotic use can generally be categorized into three main indications: empiric, targeted/definitive, and prophylactic. Prophylactic antibiotics are typically guided by the type of surgical wound (clean-contaminated or contaminated) and the likely pathogens expected during the procedure. There is ongoing debate about the necessity of prophylactic antibiotics in clean wounds, while for contaminated wounds, prophylaxis is not indicated because these wounds require treatment for an established infection. As such, obtaining a bacterial culture is not a prerequisite for surgical prophylaxis, which is specifically intended to prevent infections from anticipated pathogens rather than treat an existing one. I would suggest you revise the sentence to something like:

‘The poor performance of culture in our study could be explained by the common practice in the neonatal ward of initiating systematic empirical antibiotic therapy, prior to obtaining bacterial cultures.’

Response: We have modified the phase according to your recommendations

Recommendation 8: Lines 358-359: ‘However, differences between men and women in terms of personal hygiene practices, of comorbidities frequency and anatomy may explain this association.’

• The phrase "in terms of personal hygiene practices, of comorbidities frequency and anatomy" is cumbersome and could be streamlined for better clarity.

• The phrase "comorbidities frequency" should be "the frequency of comorbidities" for proper grammar

I would revise the sentence as ‘However, differences between men and women in terms of personal hygiene practices, the frequency of comorbidities, and anatomy may explain this association.’

Response: We have modified the phase according to your recommendations

---

## [Editor Report · Decision Letter 2]

3 Dec 2024

PONE-D-24-26487R2Risk and outcomes of healthcare-associated infections in three hospitals in Bobo Dioulasso, Burkina Faso, 2022: a longitudinal studyPLOS ONE

Dear Dr. HEMA,

Thank you for re-submitting your manuscript to PLOS ONE. After careful consideration, we feel that it has merit but does not (yet) fully meet PLOS ONE’s publication criteria as it currently stands. Therefore, we invite you to submit a revised version of the manuscript that addresses the points raised during the review process. Please find the issues that need to be resolved below. 

We look forward to receiving your revised manuscript.

Kind regards,

Luisa Anna Denkel

Academic Editor

PLOS ONE

Journal Requirements:

Additional Editor Comments:

Dear authors,

Thank you for the revision of your manuscript that increased its quality. However, before the manuscript can be accepted for publication, a few more changes are necessary. We hope for your understanding:

1. Please use the term “Enterobacterales” instead of “Enterobacteriaceae” in the entire manuscript.

2. Please always include the country when reporting the study location including the abstract. E.g. -

- […] in three hospitals in the city of Bobo Dioulasso, Burkina Faso.

- A prospective, longitudinal, multicenter study was conducted from May 1 to November 30, 2022, in two district hospitals (Do and Dafra) and the Sourô Sanou Teaching Hospital (CHUSS), Burkina Faso.

3. Most importantly, please explain the fact that LOS > 2 days was identified as risk factor. What was the reference group? This does not make any sense. According to your inclusion criteria LOS > 48h is a prerequisite for inclusion. Please clarify this issue.

4. Please use the term “risk factors” instead of “predictors” and “etiological factors” throughout the entire manuscript. Please explain in methods how the variable “Length of stay before inclusion >2 days” has been defined.

5. Table 3:

- please correct “Length stays before inclusion >2 days” into “Length of stas before inclusion >2 days. Again, what was the reference here?

- Please explain the abbreviation CHUSS in the table legend.

---

## [Author Response · Author response to Decision Letter 2]

10 Dec 2024

Responses to academic Editor

1. Please use the term “Enterobacterales” instead of “Enterobacteriaceae” in the entire manuscript.

RESPONSE: correction made

2. Please always include the country when reporting the study location including the abstract. E.g.

- […] in three hospitals in the city of Bobo Dioulasso, Burkina Faso.

- A prospective, longitudinal, multicenter study was conducted from May 1 to November 30, 2022, in two district hospitals (Do and Dafra) and the Sourô Sanou Teaching Hospital (CHUSS), Burkina Faso.

RESPONSE: correction made

3. Most importantly, please explain the fact that LOS > 2 days was identified as risk factor. What was the reference group? This does not make any sense. According to your inclusion criteria LOS > 48h is a prerequisite for inclusion. Please clarify this issue.

RESPONSE:

We agree that this section is confusing. The term "length of stay > 2 days before inclusion" refers to patients who spent more than two days in another ward in the same hospital or in another hospital before being transferred to the study ward. They become potentially eligible for our study after their transfer in one of the study wards. To be included in our study, the patient must have no signs of infection on admission and spend at least two days in one of the selected wards. Our aim was to compare patients who had spent at more than 48 hours in another ward in the same hospital or in another hospital before being admitted in the study wards (exposure group) with patients who had no such history, i.e. patients admitted directly to the study wards and those transferred to the study wards the day after admission (control group). The aim was to determine the relationship between the duration of exposure to the hospital environment and the risk of healthcare-associated infection (HAI).

We will use the term “Previous hospital stay” instead of “Length of hospital stays before inclusion >2 days”

4. Please use the term “risk factors” instead of “predictors” and “etiological factors” throughout the entire manuscript. Please explain in methods how the variable “Length of stay before inclusion >2 days” has been defined.

RESPONSE: correction made

5. Table 3:

- please correct “Length stays before inclusion >2 days” into “Length of stas before inclusion >2 days. Again, what was the reference here?

- Please explain the abbreviation CHUSS in the table legend.

RESPONSE: correction made

---

## [Editor Report · Decision Letter 3]

12 Dec 2024

Risk and outcomes of healthcare-associated infections in three hospitals in Bobo Dioulasso, Burkina Faso, 2022: a longitudinal study

PONE-D-24-26487R3

Dear Dr. Arsene HEMA,

We’re pleased to inform you that your manuscript has been judged scientifically suitable for publication and will be formally accepted for publication once it meets all outstanding technical requirements.

Kind regards,

Luisa Anna Denkel

Academic Editor

PLOS ONE

---

## [Editor Report · Acceptance letter]

PONE-D-24-26487R3

PLOS ONE

Dear Dr. HEMA,

I'm pleased to inform you that your manuscript has been deemed suitable for publication in PLOS ONE. Congratulations! Your manuscript is now being handed over to our production team.

Kind regards,

on behalf of

Dr. Luisa Anna Denkel

Academic Editor

PLOS ONE